# The Remodeling of Dermal Collagen Fibrous Structures in Mice under Zero Gravity: The Role of Mast Cells

**DOI:** 10.3390/ijms24031939

**Published:** 2023-01-18

**Authors:** Viktoriya Shishkina, Andrey Kostin, Artem Volodkin, Vera Samoilova, Igor Buchwalow, Markus Tiemann, Dmitri Atiakshin

**Affiliations:** 1Research Institute of Experimental Biology and Medicine, Burdenko Voronezh State Medical University, 394036 Voronezh, Russia; 2Research and Educational Resource Center for Immunophenotyping, Digital Spatial Profiling and Ultrastructural Analysis Innovative Technologies, Peoples’ Friendship University of Russia, 117198 Moscow, Russia; 3Institute for Hematopathology, 22547 Hamburg, Germany

**Keywords:** extracellular matrix, collagen fibers, mast cell, fibroblasts, zero gravity, skin

## Abstract

Mechanisms of adaptive rearrangements of the fibrous extracellular matrix of connective tissues under microgravity practically remain unexplored, despite the most essential functions of the stroma existing to ensure the physiological activity of internal organs. Here we analyzed the biomaterial (the skin dermis) of C57BL/6J mice from the Rodent Research-4 experiment after a long stay in space flight. The biomaterial was fixed onboard the International Space Station. It was found that weightlessness resulted in a relative increase in type III collagen-rich fibers compared to other fibrous collagens in the skin. The number of mast cells in the skin did not change, but their secretory activity increased. At the same time, co-localization of mast cells with fibroblasts, as well as impregnated fibers, was reduced. Potential molecular–cellular causes of changes in the activity of fibrillogenesis under zero-gravity conditions and the slowdown of the polymerization of tropocollagen molecules into supramolecular fibrous structures, as well as a relative decrease in the number of fibrous structures with a predominant content of type-I collagen, are discussed. The data obtained evidence of the different sensitivity levels of the fibrous and cellular components of a specific tissue microenvironment of the skin to zero-gravity conditions. The obtained data should be taken into account in the systematic planning of long-term space missions in order to improve the prevention of undesirable effects of weightlessness.

## 1. Introduction

An increased duration of space flights poses challenges for the scientific community to improve measures to prevent the adverse physiological effects of microgravity. The connective tissue, forming an integrative-buffer metabolic environment with key parameters in the specific tissue microenvironment of each organ is of paramount significance for the quality of adaptation to the conditions of altered gravity.

The musculoskeletal system demonstrates the most sensitivity to gravity; that is why changes in the musculoskeletal system, including “mechanosensory” elements of the extracellular matrix, are actively investigated by space physicians, physiologists, and biologists [1,2,3]. However, the state of intra-organ connective tissues under microgravity conditions remains practically unexplored, excluding limited digestive organs [4,5,6].

Fibrous structures of all types of connective tissue are the main structural elements that allow implementing the biomechanical function of a particular organ. Histochemical methods of light-optical microscopy, developed at the end of the 19th century, remain relevant in modern morphological studies to identify the main component of fibrillar fibers: type-I collagen. A pattern used to detect and subsequently analyze type-I and -III collagens in the fibrous component of the extracellular matrix using polarizing microscopy after staining with picrosirius red was proposed by L.C. Jungueira et al. [7]. At present, this approach is still used to identify and perform the qualitative analysis of connective tissue fibers to detect the content of type-I and -III collagens. Collagen provides strength, elasticity, and tissue histo-architectonics and participates in response to external and internal stress factors [8].

Mast cells (MCs) play an essential role in the remodeling of both fibrous elements and amorphous substances of the skin stroma [9]. The functional features of the MC secretome allow them to act as inducers and regulators of the most important physiological and pathological processes, including the coordination of tissue microenvironment homeostasis, the implementation of innate and adaptive immunity, the initiation and subsequent development of inflammatory and allergic reactions, angiogenesis, and cell proliferation. MCs are able to monitor the state of the connective tissue of a specific tissue microenvironment and participate in the remodeling of the histo-architectonics of internal organs [9,10,11,12,13]. A number of secretome components, including proteoglycans, specific proteases, and TGF-β, have direct and indirect effects on changing the stroma structure of internal organs. The frequent association of MC numbers with the intensity of fibrotic changes in organs proves the hypothesis of their close involvement in both the regulation of the activity of collagen-producing cells and extracellular stages of fibrillogenesis [9]. Thus, MCs are important participants in the formation of the biological effects of zero gravity on the stroma of various human and animal organs. There is a particular quantity of mast cells in organs that are located on the border of the external and internal environment, including the skin. However, to date, the fibrillotropic features of the mast cell population have not been adequately studied in the field of space biomedicine research. The authors experimentally studied the features of the remodeling of collagen fibrous structures of the skin dermis, including the state of mast cells, in C57BL/6J mice following a space flight to the International Space Station (ISS).

## 2. Results

The skin dermis of mice in all control groups had a well-defined collagen-elastic fibrous scaffold with a predominance of type-I collagen (Figure 1a). The predominant distribution of type-I collagen was detected throughout the fiber bundles (Figure 1a). The surface of the collagen fibers had a pronounced longitudinal pattern with a well-developed system of transverse fibrils. The presence of these zones is of great functional significance in preserving the structural integrity of the frame and maintaining its three-dimensional configuration. Type-III collagen was detected in the green spectrum in small amounts (approximately 10% of the area of all collagen fibrous structures positive for type-I and -III collagens), with a predominant location on the periphery of collagen fiber bundles. The highest amount of type-III collagen was detected in the foci of fibrillogenesis, including the distribution sites of MCs and fibroblasts (Figure 1b). The data obtained were also confirmed using combined histochemical staining with toluidine blue and silver impregnation (Figure 1c–f). Reticular fibers were rarely detected in the skin dermis of the control groups, the most pronounced in the area of mast cell adherence to fibroblasts or fibrocytes. It should be noted that in animals from the control groups, MCs were often detected in close proximity to fibroblasts and fibrocytes (Figure 2a–e). The state of the skin MC population in animals from the control groups was the same in terms of both quantitative features and secretory activity (Table 1 and Table 2). MCs located near the basement membrane of the epidermis presented smaller sizes, as well as the sizes of secretory granules.

MCs in the reticular dermis were of larger sizes and contained granules of various sizes (Figure 2a–d). Some granules could exceed 1 µm in size, representing individual, large, rounded formations (Figure 2b,c,e–h). The number of non-degranulated forms of MCs was about a quarter of the entire population (Table 3). MCs with morphological signs of granule secretion predominated in the population (Figure 2a–e). Secretory granules were detected at a considerable distance from MCs (Figure 2a–d). Sometimes, MCs formed numerous groups, in which they, as a rule, were co-localized with representatives of fibroblastic differons (Figure 2a).

The 21–24-day stay of mice in microgravity conditions was accompanied by particular changes occurring in the arrangement of the specific tissue microenvironment of the skin. A certain “looseness” of the reticular layer of the dermis was visually noted due to the expansion of the interfascicular spaces (Figure 1g). Notably, the papillary dermis contained short and slightly convoluted fibers forming a cellular structure (Figure 1g). In the collagen fibrous component of the connective tissue of the skin dermis, the relative content of type-III to -I collagen increased almost two times compared with animals from the control groups and reached 19.2%. Concurrently, the detection of type-III collagen in the fibers of the mast cell microenvironment decreased (Figure 1h). A similar pattern was observed after the use of combined histochemical staining with toluidine blue and silver impregnation, demonstrating the almost complete disappearance of reticular fibers around the mast cells (Figure 1i–l). Moreover, it should be noted that even mast cell degranulation was not accompanied by an increased intensity of the formation of impregnated fibers (Figure 2l).

These morphological signs evidencing the gravisensitivity of the fibrous extracellular matrix of skin connective tissue remodeling were combined with a number of changes in the state of the MC population (Table 2 and Table 3). Despite the fact that the space flight conditions did not result in a significant change in the population size of skin MCs (Table 2), the secretory profile of MCs significantly changed in zero-gravity conditions with an increased pool of degranulating cells compared to the control groups (Table 3).

A relatively distinct feature of the space flight group was a significant increase in the area of particular MCs, a change in their shape with the formation of cytoplasmic outgrowths at a considerable distance, and the detection of secretory granules at a distance from the cytoplasm, with the formation of metachromatic loci in the skin dermis (Figure 2i–n). In general, the intensity of the release of mast cell secretome components in the specific tissue microenvironment increased. In addition, the granules were freely located in the extracellular space over a large area (Figure 1j and Figure 2i,j).

In terms of packaging the secretory material, it is necessary to note the presence of secretory granules of a smaller size in the MC cytoplasm in animals onboard the ISS compared to animals from the control groups, with a significant decrease in the number of large metachromatic granules, larger than 1 μm. Concurrently, the heterogeneity of granule sizes was significantly reduced. On the one hand, this may indicate an active consumption of biogenesis products and a decrease in the reserves of preformed mediators for adaptation to changed gravity. However, on the other hand, this fact may also evidence specific features of the granule metabolism under microgravity conditions.

Under space flight conditions, there was a change in the frequency of MC co-localization with fibroblastic differon cells. In particular, the adherence of MCs to fibroblasts and fibrocytes was detected less frequently in the skin of animals from the space flight group compared to animals of basal, vivarium, and ground controls (Figure 1j–l and Figure 2j–l).

The dissociation of the morphofunctional cluster “mast cell–fibroblast” became a specific morphological feature of the space flight group in terms of fibrillogenesis effectiveness, as evidenced by the absence of impregnated fibers in the pericellular microenvironment of mast cells located near fibroblasts, despite active secretion. In animals from the space flight group, despite the existing individual differences, one can note a rarer detection of fibers containing type-III collagen (Figure 1h) or impregnated fibers (Figure 1i–l) compared with the data obtained when studying the biomaterial of the control groups. Even secretory granules’ co-localization with fine fibers did not lead to their impregnation, a specific feature of reticular fibers (Figure 1i). Despite this, the intensity of liberalization of the MC secretome components in the specific tissue microenvironment of the skin increased under microgravity conditions (Table 1, Figure 1i–l and Figure 2i–n). Notably, there was an impression of a “looser” arrangement of secretory granules in the MC cytoplasm, which could occupy a large area in the skin dermis (Figure 2i,j).

## 3. Discussion

The study of fibrillogenesis as a fundamental process under zero-gravity conditions allows for the discovery of novel mechanisms of extracellular matrix remodeling. A more incisive understanding of the molecular mechanisms of the stromal renewal of internal organs provides novel algorithms for connective tissue remodeling under both physiological and pathological conditions. This determines the importance of developing methodological approaches that help specify the points of application of MC selective effects in the process of collagen fiber formation and their introduction into morphological practice, as well as a more unbiased evaluation of their contribution to the development of pathological processes with progressive tissue fibrosis [9].

Discussing the results obtained, it should be noted that there is a certain gravisensitivity of the connective tissue of internal organs, the fact being demonstrated in the studies of the digestive system previously performed [4,5,6]. Similar results were obtained in an experiment conducted on Japanese quail chicks under orbital flight conditions onboard the Mir space station [14].

Collagen is the main polymer in all types of connective tissues, ranging from fibrous to skeletal tissues. Collagen networks provide an organ with strength and resistance when mechanical force is applied, thereby preventing excessive tissue deformation [8]. At present, there are 29 types of collagen known, of these, types I, II, and III account for almost 80% of collagen present in the body [8,15]. Intertwining collagen fibers form long, fine fibrils. Other types of collagens bind to fibrillar-type collagen. Various types of collagens, together with other components of the extracellular matrix, undergo constant remodeling to achieve the required strength and elasticity under biomechanical loading conditions. Physical, mechanical, or chemical damage to a tissue or organ will result in impaired collagen accumulation and arrangement. Therefore, the assessment of collagen distribution patterns provides an idea of the tissue’s and organ’s resistance to stretching [8].

Picrosirius red (also called PSR or sirius red) staining is a selective histochemical technique suitable for both the morphological and semi-quantitative detection of collagen fibers in paraffin sections. Sections analyzed by polarizing light microscopy (PLM) simultaneously presented a complex mixture of different colors labeling different types of collagen, which allowed us to identify and analyze the differential distribution patterns of structurally different types of collagen in tissues. In addition, PLM can be performed using a light microscope and adding two filters, a polarizer, and an analyzer, which is a powerful and reliable tool to assess the properties of fibers in tissues containing collagen fibers [16].

Picrosirius red specifically stains type-I and -III collagens, which provides the potential to use this staining technique to isolate and quantify liver [17] and heart fibroses [18], to identify collagen fibers in paraffin sections of the colon [19] and oral cavity organs [20] in normal and inflammatory conditions, in the study of articular cartilage [21] and collagen in the pulmonary artery wall [22], glomerulopathologies [23], atherosclerosis [24], cancer [25], etc. The sulfonic acid group of picrosirius red reacts with the basic amino groups of lysine, hydroxylysine, and the guanidine group of arginine, which is present in the collagen molecule [26]. As an anionic dye, picrosirius red binds to all various isoforms of collagen. In the light field, collagen is visualized as bundles of pink or red fibers, the architectonics of which change under pathological conditions [26]. In polarizing microscopy, larger collagen fibers appear bright yellow or orange, while finer ones, including reticular fibers, appear green. Birefringence (double refraction), in which incident light is split into two different paths by polarization, is very specific to collagen. The amount of polarized light absorbed by the picrosirius red dye strictly depends on the orientation of the collagen fibrils, which allows differentiating different types of collagens [27]. Our proposed variants of histochemical protocols for combined staining with Giemsa solution with picrosirius red and silver impregnation can be used to interpret the involvement of mast cells in the remodeling of the intercellular matrix of the tissue microenvironment during the development of adaptive and pathological processes [9].

A decreased number of pre-collagen fibers adjacent to MCs or extending from them in different directions of the extracellular matrix was a significant morphological sign evidencing the weakening activity of MC participation in fibrillogenesis under zero-gravity conditions. This possibly resulted in an integral decrease in the number of impregnated fibers in the skin dermis in mice from the space flight group compared with the animals of the control groups, supporting the fact of fibrillogenesis inhibition under zero-gravity conditions. Attention was drawn to the fact of more frequent MC co-localization with collagen fibers, which may indicate a change in their migration activity within the specific tissue microenvironment of the skin.

On the other hand, the integral increase in collagen fibers containing type-III collagen demonstrates that, along with the weakening formation of reticular fibers in the local microenvironment of fibroblasts and mast cells, the gravity-induced transformation of mature collagen fibers is actively implemented. However, the mechanisms of these changes remain unclear, suggesting the participation of enzymes capable of destroying type-I collagen, including matrix metalloproteinases.

The long-term stay of secretory granules in the extracellular matrix allows for the additional control of the metabolic reaction intensity of a specific tissue microenvironment and the state of the extracellular matrix. Specific MC proteases have pronounced properties for the direct and indirect remodeling of connective tissue fibrous structures [9], which, under hypogravity conditions, provides MCs with special properties.

The analysis of the morphofunctional cluster “mast cell–fibroblast” is an important axis of fibrillogenesis. In recent years, it has been noted that in addition to the mast cell secretome, which consists of substances that activate fibroblasts, the direct adhesion between these two cell types is also a pre-requisite for the activation of fibrillogenesis [9]. Thus, the adherence of MCs to fibroblasts may indicate certain potentials in changing the equilibrium state, in particular, in initiating the process of fibrillogenesis. MC localization near the fibers may support the participation of proteoglycans secreted by these MCs in the fibril thickening process. MC degranulation may be accompanied by the achievement of the required concentration of signaling and structural molecules within a strictly limited microlocus [9]. In addition, MC granules can act as “nucleators”—points (molecular loci) initiating the start of polymerization of collagen molecules and fibrillogenesis onset [9].

Concurrently, the effect of a particularly increased integral level of type-III collagen detected in the skin dermis according to the results of picrosirius red staining may evidence the development of different mechanisms of extracellular matrix remodeling in zero-gravity conditions, leading to the certain disorganization of fibrous structures containing type-I collagen. An increased activity of the MC-mediated secretion of specific matrix metalloproteinases, visually separated from fibroblasts, can be assumed as one of these mechanisms. These include MMP8 and MMP13, which have a high potential to initiate the degradation of type-I triple-helix collagen [28].

Microgravity conditions can apparently result in alterations of fibrillogenesis conditions. In addition to the changes in the gravitational stimulus, the same has been reported about the development of hemodynamic changes causing additional modifications of the integrative-buffer metabolic environment.

A decreased efficiency of the extracellular assembly of fibrous collagens in the intercellular matrix may be due to a change in the pH level, the content of proteins in the amorphous phase of the extracellular matrix, water, etc. Thus, the formation of a collagen fiber will occur under different conditions of water content, concentration of tropocollagen and other proteins, osmotic pressure, ionic strength, and other factors. It should be taken into account that in case of skin damage resulting from the professional activity of astronauts onboard the ISS, processes of fibrillogenesis may not occur in full due to gravitation-induced changes in the parameters of the local tissue microenvironment.

The obtained morphological signs evidence the gravisensitivity of the fibrous components of the skin dermis and the “key” cellular initiators of fibrillogenesis. The orbital flight setting led to the MCs’ dissociation with representatives of fibroblastic differon, which supports alterations in the state of fibrillogenesis under normal conditions of gravitational loading. An alteration in the gravitational signal results in alterations in hemodynamic conditions—pH—playing an important role in the process of fibrillogenesis and protein content, thus causing the restructuring of the specific tissue microenvironment. Zero-gravity conditions and the duration of a space flight form the adaptive settings of organs containing the connective tissue in their composition. The mechanisms of the physiological regeneration of the fibrous connective tissue cannot be implemented in full, which should be taken into account when considering the professional activity of astronauts onboard the ISS.

## 4. Materials and Methods

### 4.1. Experimental Design

The Rodent Research-4 experiment was performed on C57BL/6J *Mus musculus* mice, aged 9–12 weeks old, in 2017; the animals were delivered from the Jackson Laboratory (Jackson Laboratory, Bar Harbor, ME, USA). The main object for morphological research was the skin of the lateral surface of the thigh. The study included 40 mice (male) divided into 4 groups: the space flight group (n = 10) and three control groups containing ground (n = 10), basal (n = 10), and vivarium (n = 10) animals (Table 3). The animals from the space flight group were kept in a transport container in the Dragon SV capsule on the launch pad for 4 days since 18 February 2017. The SpaceX-10 spacecraft was launched on 23 February, laboratory mice were delivered to the ISS on 25 February, and stayed in space for 21 to 24 days. The unique feature of the biomaterial of the space flight group was the fact that euthanasia was performed directly under zero-gravity conditions onboard the ISS. The mice from the vivarium control group received drinking water and food ad libitum. The mice from the basal control group were euthanized immediately after the launch of the SpaceX-10 spacecraft. The mice from the ground control group were kept in the ground experimental chamber simulating conditions onboard the ISS (John F. Kennedy Space Center, Merritt Island, FL, USA); the duration of stay was similar to that of the space flight experiment. Samples were frozen and then kept in ice packs. The time interval between euthanasia and cryofixation of the biomaterial was 2 min. Biomaterial (before landing) was stored in a freezer. The protocol of work with animals from the control groups corresponded to the schedule of work with animals from the space flight group.

The obtained samples, according to the NASA–Roscosmos protocol “Utilization Sharing Plan On-Board ISS” (signed on 18 July 2013), were delivered to the Russian Federation in dry ice without defrosting, which was controlled by temperature detectors. All animal procedures were approved by the Ames Institution Animal Care and Use Committee (protocol CAS-13-001-Y1, approved 13 May 2014, Ames, IA, USA).

### 4.2. Histoprocessing

After fragile defrosting, skin dermal samples were fixed in 10% buffered formaldehyde for 12 h at 4 °C. Then, histological processing was conducted according to the standard protocol using a number of alcohols, xylene, and paraffin in an MTP-120 carousel tissue processor (SLEE Medical, Nieder-Olm, Germany), followed by the preparation of paraffin blocks at the Tissue-Tek TEC 5 Embedding Station (Sakura Seiki Co., Ltd., Nagano, Japan). Histological sections 4 µm thick were created on Accu-Cut SRM 200 Rotary Microtome (Sakura Finetek Europe B.V, Alphen aan den Rijn, The Netherlands).

### 4.3. Tissue Probe Staining

Histological skin sections were impregnated with silver and stained with picrosirius red, toluidine blue, and Giemsa’s solution [29]. Histochemical techniques in which Giemsa staining was combined with silver impregnation or picrosirius red staining were used to simultaneously detect mast cells and the fibrous component [9].

### 4.4. Image Acquisition

Stained tissue sections were observed on a ZEISS Axio Imager.A2 equipped with a Zeiss alpha Plan-Apochromat objective 100×/1.46 Oil DIC (UV) VIS-IR and AxioCam digital microscope cameras (Axiocam 506 color and Axiocam 503 monochrome CCD). Captured images were processed with the software program ZEN 2.3 (Carl Zeiss Vision, Jena, Germany) and submitted with the final revision of the manuscript at 300 DPI. The volume of the MC population was determined in conditional fields of view using a ×40 objective; the number of fields of view was at least 50 to obtain a representative data array. After conducting the planimetric analysis, in order to facilitate the perception of the resulting digital array, the results were adapted to the 1 mm^2^ area of skin tissue.

The ratio of various collagens in the fibrous structures of the connective tissue of the skin was studied in polarized light after staining with picrosirius red, in which type-I collagen was visualized in yellow-orange (red) shades, and type-III collagen in green (Figure 1a,b,g,h) [30]. The ratios of type-I and -III collagens were determined using ImageJ software by analyzing the color histogram of each visual field. The absolute parameters of the red and green colors of the spectrum were converted into relative values and expressed in percentage correlations relative to each other, considering the standard deviation.

### 4.5. Statistical Analysis

Statistical analysis was performed using the SPSS software package (Version 13.0). The results are presented as mean (M) ± m (standard error of the mean). To assess the significance of the differences between the two groups, Student’s *t*-test or Mann–Whitney U test in the case of a nonparametric distribution was used.

## 5. Conclusions

The results obtained indicate the existence of a certain gravisensitivity of the stroma of internal organs, which must be taken into account in space biomedicine in order to improve the prevention of undesirable effects of weightlessness and preserve the health of astronauts during flights in near and deep space. Changes in the specific tissue microenvironment of the skin during orbital flight can affect the mechanisms of remodeling of the extracellular matrix of the dermis involving mast cells, as well as the functioning of the immune landscape providing the protective function of the skin.

## Figures and Tables

**Figure 1 ijms-24-01939-f001:**
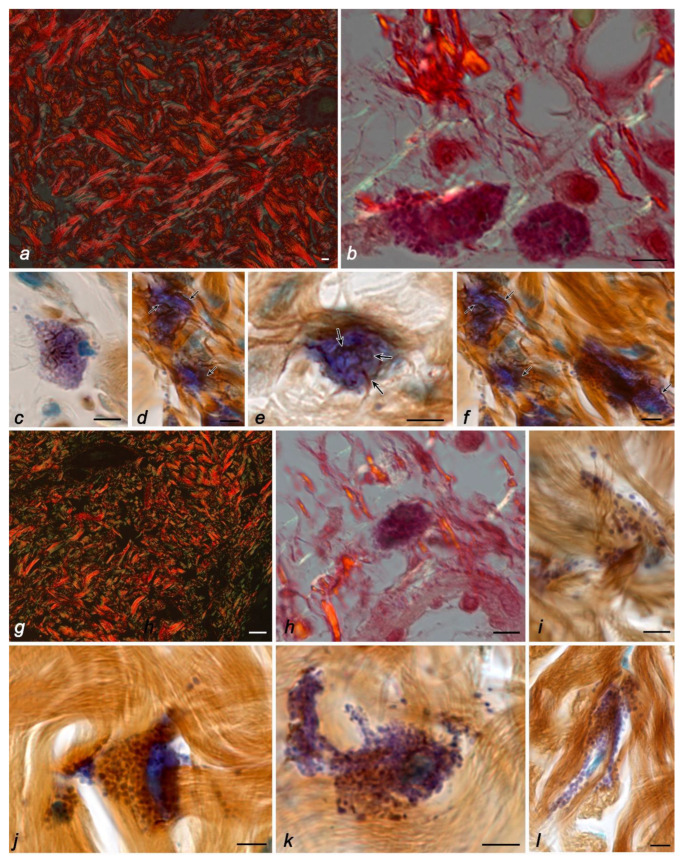
The fibrous component of the extracellular matrix and mast cells in the skin of the thigh of C57BL/6J mice. Groups of vivarium control (**a**,**c**,**d**,**f**), ground control (**b**,**e**), and space flight (**g**–**l**). Techniques: staining with picrosirius red (**a**,**g**), picrosirius red, and Giemsa solution (**b**,**h**), impregnation with silver and toluidine blue (**c**–**f**,**i**–**l**). Notes**:** (**a**) visualization of the dermis in polarized light: fibers containing type-I (fluorescently labeled red regions) and -III collagens (fluorescently labeled green regions). (**b**) Fibers containing type-III collagen (fluorescently labeled green regions, indicated by an arrow) in the local microenvironment of mast cells (indicated by a double arrow), fibers containing type-I collagen are red. (**c**–**f**) Different distributions of fine precollagen fibers (impregnated, indicated by arrow) in the local mast cell microenvironment. (**g**) Visualization of the dermis in polarized light: fibers containing type-I (fluorescently labeled red regions) and -III collagens (fluorescently labeled green regions). (**h**) In the local microenvironment of the mast cell (indicated by a double arrow), fibers containing type-I collagen predominate (fluorescently labeled red regions, indicated by an arrow), fibers containing type-III collagen are green. (**i**–**l**) Lack of impregnated fibers in the local mast cell microenvironment. Scale bar: (**a**,**g**) 20 µm; the rest 5 µm.

**Figure 2 ijms-24-01939-f002:**
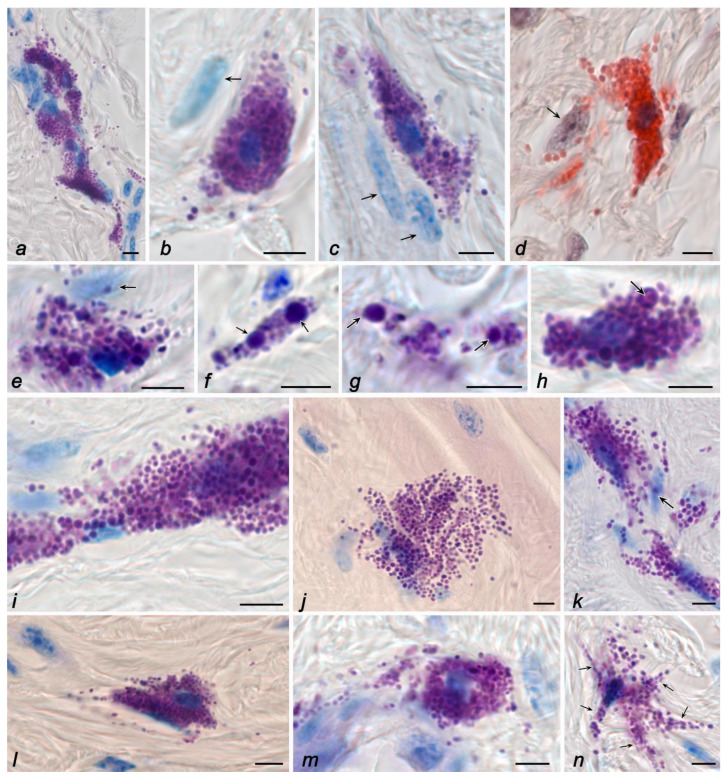
The mast cells in the skin of the thigh of C57BL/6J mice. Groups of basal (**a**,**d**,**h**), vivarium (**b**,**g**), and ground (**c**,**e**,**f**) controls, and space flight (**i**–**n**). Techniques: staining with toluidine blue (**a**–**c**,**e**–**i**,**k**,**m**,**n**), Giemsa solution (**j**,**l**), and detection of chloroacetate esterase activity (**d**). Notes: (**a**) Fibroblast-associated mast cells form a multicellular fibrillogenic cluster. Active excretion of mast cell secretome into the extracellular matrix. (**b**–**e**) Co-localization of mast cells and secretory granules with fibroblasts (indicated by an arrow). (**f**,**g**) Location of secretory granules in the extracellular matrix of the connective tissue of the skin dermis, including large-sized granules (indicated by an arrow). (**h**) Loss of secretome metachromacy in the central region of a large mast cell granule during secretion (indicated by an arrow). (**i**,**k**) Increased secretory activity of contacting mast cells. (**j**) Large area of mast cell cytoplasm, diffuse distribution of granules in the reticular dermis. (**l**,**m**) Active secretion of mast cells in the reticular dermis with distribution of granules in the specific tissue microenvironment of the skin. (**n**) Formation of mast cells with outgrowths of the cytoplasm (indicated by an arrow). Scale bar: 5 µm.

**Table 1 ijms-24-01939-t001:** Features of the skin MC population in mice (Technique: staining with Giemsa solution).

Groups of the Experiment	Number of MCs (Per 1 mm^2^)	The Frequency of MCs’ Adherence to Each Other (in % of the Total Number)
Vivarium control	34.4 ± 3.1	4.59 ± 0.29 ∞
Basal control	39.1 ± 4.6	5.24 ± 0.41 ∞ *
Ground control	32.6 ± 3.2	3.75 ± 0.27 +
Space flight	36.6 ± 3.2	3.51 ± 0.18 + ⸋

Notes: Significant differences (*p* < 0.05) compared with the group of ground control (*), basal control (+), vivarium control (⸋), and space flight (∞).

**Table 2 ijms-24-01939-t002:** Secretory activity of the skin MC population in mice, % (Technique: staining with Giemsa solution).

Groups of the Experiment	MCs without Signs of Secretion	MCs with Signs of Secretion	Nuclear-Free Fragments of MCs
Vivarium control	26.2 ± 2.7 ∞	45.7 ± 3.7 ∞ +	28.1 ± 1.9
Basal control	21.8 ± 2.4 ∞	54.8 ± 2.1 ⸋ ∞	23.4 ± 2.5
Ground control	25.3 ± 2.1 ∞	47.1 ± 4.1 ∞	27.6 ± 2.1
Space flight	11.5 ± 0.8 * + ⸋	61.5 ± 3.3 * + ⸋	27.0 ± 1.8

Notes: Significant differences (*p* < 0.05) compared with the group of ground control (*), basal control (+), vivarium control (⸋), and space flight (∞).

**Table 3 ijms-24-01939-t003:** Reagents used for histochemical staining of mouse skin.

Dyes	Catalogue Number	Provider	Dilution	Manufacturer
Toluidine blue	07-002	Biovitrum	Ready-to-use	ErgoProduction LLC, Saint-Petersburg, Russia
Giemsa solution	20-043/L	Biovitrum	Ready-to-use	ErgoProduction LLC, Russia
Picrosirius red	ab150681—Picro Sirius Red Stain Kit (Connective Tissue Stain)	Abcam	Ready-to-use	Abcam, Cambridge, UK
Silver impregnation	21-026	Biovitrum	Ready-to-use	ErgoProduction LLC, Russia

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
