# Peer review of "The Remodeling of Dermal Collagen Fibrous Structures in Mice under Zero Gravity: The Role of Mast Cells"

_ijms, 2023, doi:10.3390/ijms24031939_

Round 1
Reviewer 1 Report
Shishkina et. al. addressed the impact of zero gravity conditions over tissue remodeling, specifically on musculoskeletal tissue on thighs from mice. Although, these findings are relevant in the field, several suggestions/questions arose.
Authors did not spell in the manuscript the acronym ISS used in lines 16 and 72.
Line 92, it is not clear, at least to me, if the authors mean that the granule size is also enlarged (after reading the manuscript this is clarified).
Table 1 should be shown complete, it is cut in line 93
It is very distracting the way figure 1 and figure 2 are described. Images from same groups should be clustered regardless of the staining, this will make it easier to understand for the readers.
Line 181, the active secretion of fibroblast-associated MCs..this must be re- written
Lines 163 and 186, authors used excretion, but secretion or released are more proper terms according to the context.
Line 292, There is a different reference format used.
In regards of section discussion, it seems very extensive. For instance, paragraphs 265-272 and 273-304 described biogenesis of collagen rather than discussing the findings. These paragraphs should be deleted and include paragraph 317-324 right after the line 259, which is addressing the potential role of MMPs.
Discussion could be improved if authors speculate/discuss, for instance, for how long these effects may last (additional groups must have been included, in which different time points after removing from zero gravity, stainings could be done).
Authors did not mention mice sex and this could also be discussed.
Manuscript structure is altered, conclusions section should be after discussion.
Author Response
To Reviewer #1
Reviewer: Shishkina et. al. addressed the impact of zero gravity conditions over tissue remodeling, specifically on musculoskeletal tissue on thighs from mice. Although, these findings are relevant in the field, several suggestions/questions arose.
Authors: The authors thank the Reviewer for the careful reading of the manuscript and for his valuable comments, which were correspondingly taken into account.
Reviewer:
1.Authors did not spell in the manuscript the acronym ISS used in lines 16 and 72.
Corrected.
2.Line 92, it is not clear, at least to me, if the authors mean that the granule size is also enlarged (after reading the manuscript this is clarified).
Corrected.
3.Table 1 should be shown complete, it is cut in line 93
Corrected.
4.It is very distracting the way figure 1 and figure 2 are described. Images from same groups should be clustered regardless of the staining, this will make it easier to understand for the readers.
5.Line 181, the active secretion of fibroblast-associated MCs..this must be re- written
Corrected.
6.Lines 163 and 186, authors used excretion, but secretion or released are more proper terms according to the context.
Corrected.
7.Line 292, There is a different reference format used.
Corrected. This literature discussion block was excluded on the recommendation of the reviewer).
8.In regards of section discussion, it seems very extensive. For instance, paragraphs 265-272 and 273-304 described biogenesis of collagen rather than discussing the findings. These paragraphs should be deleted and include paragraph 317-324 right after the line 259, which is addressing the potential role of MMPs.
Corrected.
9.Discussion could be improved if authors speculate/discuss, for instance, for how long these effects may last (additional groups must have been included, in which different time points after removing from zero gravity, stainings could be done).
Authors: In accordance with objective reasons - the program of the scientific experiment must be coordinated with the ISS crew change schedule, which was not done during the RR4 experiment. In addition, any descent from orbit implies the impact of many factors, including stress, overload during descent from orbit, impact when the spacecraft lands on the earth's surface, stay for several hours (up to a day) after landing in terrestrial conditions until the biomaterial is taken, etc. Therefore, the exclusive advantage of this experiment is the fixation of the biomaterial directly in weightlessness, which provides the most relevant information about the biological effects of weightlessness.
- Authors did not mention mice sex and this could also be discussed.
Corrected in «4.1. Experimental design».
- Manuscript structure is altered; conclusions section should be after discussion.
Corrected.
Thank you,
Igor Buchwalow

Reviewer 2 Report
This study is a very interesting research topic.
For a better manuscript, please consider the following
1) Immune cells that affect fibrillogenesis at zero gravity, other than mast cells, would be good to insert if there is a previous study.
2) Have you ever observed the change of elastin according to the change of gravity?
3) 8 page 292 line, is it correct for reference mark in the manuscript ?
Author Response
To Reviewer #2
Reviewer: This study is a very interesting research topic.
Authors: The authors thank the Reviewer for the careful reading of the manuscript and for his valuable comments, which were correspondingly taken into account.
Reviewer: Immune cells that affect fibrillogenesis at zero gravity, other than mast cells, would be good to insert if there is a previous study.
Authors: Thanks to the Reviewer for the suggestion. However, unfortunately, fibrillogenesis has not been previously studied in real space experiments. We will try to continue this project and independently investigate other immunocompetent cells in the skin in future works.
Reviewer: Have you ever observed the change of elastin according to the change of gravity?
Authors: This is a very important question for space biology. We will definitely consider this issue in continuing this project. Data that we obtained in other experiments, in particular, on Mongolian gerbils (they were 12 days in orbit on the spacecraft "Photon-M3" in 2007), as well as mice (which were 30 days on board the biological satellite "BION -M1" in 2013) indicate certain changes in the elastic fibers in the composition of the blood vessels of the gastrointestinal tract.
However, it is difficult to evaluate the results obtained due to the impact on animals of many factors during landing, in particular, stress, G-forces during descent from orbit, impact during landing of the spacecraft on the surface of the earth, and also stay from 9 to 24 hours after landing in terrestrial conditions until the biomaterial was taken at the Moscow Institute of Biomedical Problems.
Reviewer: 8 page 292 line, is it correct for reference mark in the manuscript?
Authors: We have excluded this section on the advice of the Reviewer #1.
Thank you.
Igor Buchwalow

Round 2
Reviewer 1 Report
The authors have substantially improved the manuscript